# PARCON: NOISE-ROBUST COLLABORATIVE PERCEPTION VIA MULTI-MODULE PARALLEL CONNECTION

## ABSTRACT

In this paper, we investigate improving the perception performance of autonomous vehicles through communication with other vehicles and road infrastructures. To this end, we introduce a novel collaborative perception architecture, called **ParCon**, which connects multiple modules in parallel, as opposed to the sequential connections used in most other collaborative perception methods. Through extensive experiments, we demonstrate that ParCon inherits the advantages of parallel connection. Specifically, ParCon is robust to noise, as the parallel architecture allows each module to manage noise independently and complement the limitations of other modules. As a result, ParCon achieves state-of-the-art accuracy, particularly in noisy environments, such as real-world datasets, increasing detection accuracy by 6.91%. Additionally, ParCon is computationally efficient, reducing floating-point operations (FLOPs) by 11.46%.

## 1 INTRODUCTION

One of the fundamental components of autonomous vehicles (AVs) is the ability to perceive various driving environments. With the advance of deep learning, perception systems of AVs have demonstrated effectiveness in various studies, including object detection (Liu et al. [2023]; Hu et al. [2023]; Kumar et al. [2024]) and segmentation (Zhang et al. [2023]; Xu et al. [2023]; Wu et al. [2024]). However, the perception of a single vehicle alone still has limitations caused by occlusion and limited sensor range.

To overcome these, multi-agent collaborative perception has been pivotal in enhancing perception across various environments. In particular, V2X (Vehicle-to-Everything) communications have enabled collaborative perception among heterogeneous agents such as vehicles and road infrastructure. Recognized methodologies, such as CoBEVT (Xu et al. [2022a]), V2X-ViT (Xu et al. [2022b]), and Where2comm (Hu et al. [2022]), have significantly improved the performance of 3D object detection.

One of the major challenges in collaborative perception is that the ego agent must communicate with surrounding agents, inevitably introducing noise during communication. However, most V2X collaborative perception models employ a sequential architecture to connect agent-wise and spatial-wise modules (see Figure 1 (a)), which tends to amplify the impact of noise as it propagates through the entire process. To solve this problem, *parallel* architectures have been adopted in various fields (Kim et al. [2018]; Kang et al. [2023]). When applied to V2X systems, parallel architectures offer significant benefits, including improved robustness to communication noise and enhanced performance. To leverage these advantages, we propose **ParCon**, a novel V2X collaborative perception model for 3D object detection that features parallel connections as shown in Figure 1 (b).

In this paper, we introduce our novel model, ParCon, and present extensive simulation results on various collaborative perception datasets. We demonstrate that ParCon outperforms other state-of-the-art (SOTA) approaches in detection accuracy and exhibits substantially increased robustness to communication noise (see Figure 1 (c), (d)). In particular, this robustness enables the model to achieve significantly improved detection

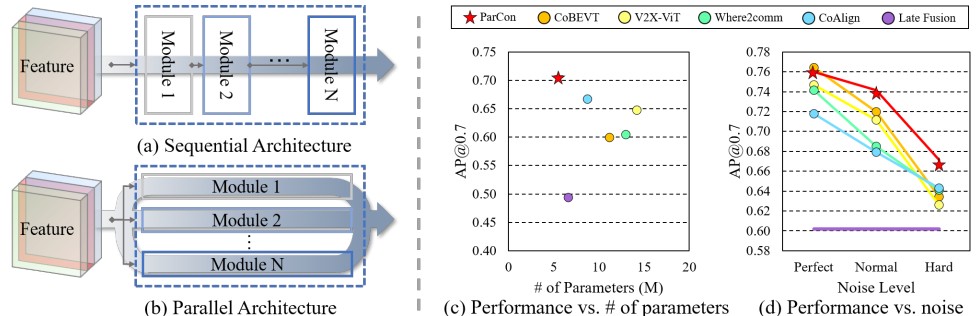

Figure 1: **About this paper.** (a) and (b) present two architectures. (c) and (d) ParCon holds state-of-the-art performance with a reduced number of model parameters and under various noises (details in Section 4.2)

accuracy in noisy environments, including real-world datasets. Furthermore, ParCon uses fewer parameters and is more computationally efficient than other SOTA approaches (see Figure 1 (c)).

We summarize our contributions as follows.

- We propose a novel *parallel* connection-based collaborative perception model, called **ParCon**, for 3D object detection. Our model achieves state-of-the-art accuracy, computation efficiency, and robustness to noise.
- We demonstrate the advantages of parallel connection. Namely, we highlight the robustness of our model to communication noise by analyzing it at different levels of communication latency, heading errors, and localization errors.
- We utilize a channel compression layer (CCL) to fuse various types of information, including agent-wise and spatial-wise, global and local information, and make the overall fusion module lightweight, thereby significantly enhancing model efficiency.

## 2 RELATED WORK

**Collaborative Perception.** Intermediate fusion, which fuses compressed features rather than raw sensor data, is the primary technique enabling collaborative perception to balance performance with communication demands. Previous studies obtain and share information using knowledge distillation (Mehr et al. [2019]), multi-module (Xu et al. [2022a]; Xu et al. [2022b]; Yang et al. [2023]), and spatial confidence-aware communication (Hu et al. [2022]). Also, to reduce the impact of noise, many studies have attempted to develop robust models, such as V2Vnet(Wang et al. [2020]), V2X-ViT(Xu et al. [2022b]), FeaCo(Gu et al. [2023]), and CoAlign(Lu et al. [2023]). To the best of our knowledge, these models fuse the information in series, and the potential of using a parallel connection has not been explored. In this paper, we design a new collaborative perception model that connects the information in parallel to enhance model efficiency and noise robustness.

**Computer vision.** The introduction of convolutional layers has revolutionized the field of computer vision; for example, AlexNet (Krizhevsky et al. [2012]), VggNet (Simonyan & Zisserman [2014]), ResNet (He et al. [2016]), etc. Convolutional layers effectively capture local features while requiring fewer parameters than fully connected layers. However, they struggle to capture relationships between distant features. Vision Transformer (ViT) (Dosovitskiy et al. [2020]) leverages self-attention mechanisms to effectively capture global relationships, but at the cost of high computational complexity and challenges in maintaining translation equivariance. Several approaches, such as Swin Transformer (Liu et al. [2021]) and Neighborhood Attention Transformer (Hassani et al. [2023]), have been introduced to solve these challenges. Similar to these approaches, our approach fully exploits the strengths of both convolution and transformer architectures to capture local features and global context, respectively.

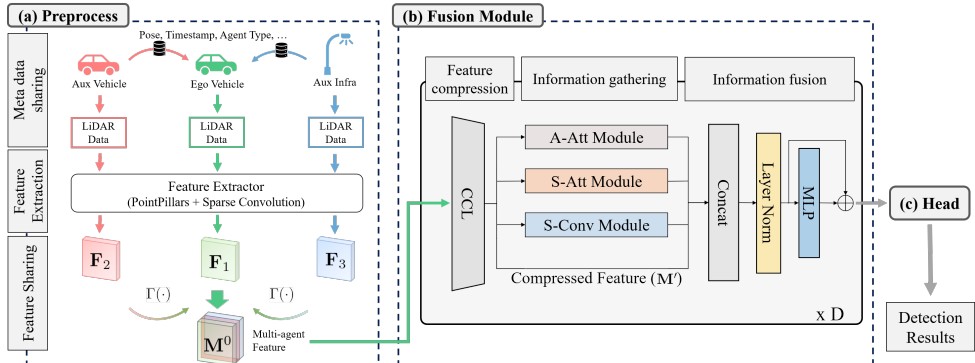

Figure 2: **Overview of our proposed collaborative perception system.** Our model consists of five steps: metadata sharing, feature extraction, feature sharing, fusion module, and detection head. The details of each component are discussed in Section 3.

## 3 METHOD

The overall architecture of ParCon is shown in Figure 2. In this section, we introduce the five main components: 1) metadata sharing, 2) feature extraction, 3) feature sharing, 4) fusion module, and 5) detection head.

### 3.1 METADATA SHARING

The *ego agent* is the center agent that performs the object detection tasks, and the *aux agents* are auxiliary agents communicating to the ego agent. Let the number of connected agents, including the ego agent, be $L$, and the ego vehicle always has an index of 1. At metadata sharing, each $l$-th aux agent for $l \in \{2, ..., L\}$ sends their agent type $t_l \in \{I, V\}$, timestamp, and pose to the ego agent. Here, the agent type I refers to an infrastructure and V to a vehicle.

### 3.2 FEATURE EXTRACTION

Most V2X collaborative perception models are based on intermediate fusion, which shares features extracted from raw point-cloud data. In ParCon, the feature extraction mainly consists of 1) PointPillars and 2) Sparse Resnet Backbone.

**PointPillars.** PointPillars (Lang et al. [2019]) splits point clouds into vertical columns. This enables PointPillars to use less memory and become faster than other voxel-based approaches, such as (Zhou & Tuzel [2018]). Also, the results of PointPillars are 2D pseudo-images. Thus, it is proper to apply a 2D convolution layer. To achieve efficient extraction, we employ PointPillars to convert point clouds of all agents to 2D pseudo-images.

**Sparse Resnet Backbone.** Sparsity is an important property of point clouds. The works (Graham et al. [2018]; Yan et al. [2018]) suggest sub-manifold sparse convolution and sparse convolution, both of which are more effective and efficient for extracting features from sparse data. They extract feature $\mathbf{F}_l \in \mathbb{R}^{H \times W \times C}$ from the $l$-th agent's 2D pseudo-image where $H$ is the height, $W$ the width, and $C$ the channels. All agents use the same backbone parameters. We deploy the Sparse Resnet (SpRes) backbone (Shi et al. [2022]; Yin et al. [2021]; Zhu et al. [2019]), which reduces memory usage and works more effectively than conventional dense convolution.

### 3.3 FEATURE SHARING

The ego agent receives the aux agents' features through communication. The features are first compressed to satisfy communication bandwidth and latency. Then, the ego agent receives the compressed features, decompresses them, transforms them into the ego agent coordinate, crops unnecessary information, and compensates for the time delay.

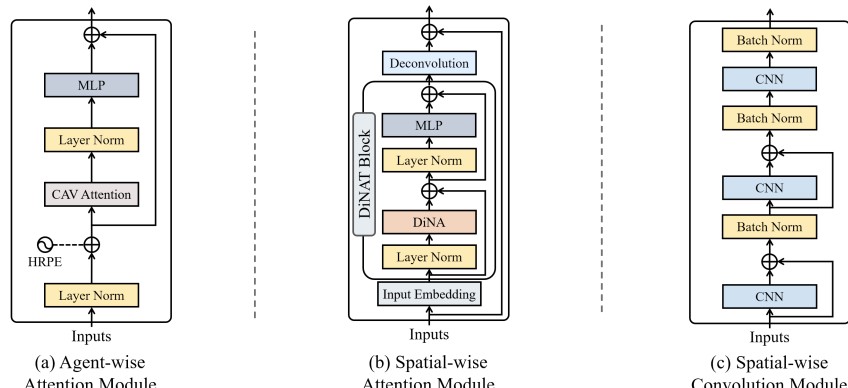

Figure 3: **Sub-modules in ParCon.** (a) Agent-wise Attention (A-Att) sub-module with HRPE. (b) Spatial-wise Attention (S-Att) sub-module. (c) Spatial-wise Convolution (S-Conv) sub-module. These modules are discussed in Section 3.4.

**Compression and Decompression.** For feature compression, all agents use the same encoder and decoder parameters. After an aux agent compresses its feature and sends it to the ego agent, the ego agent restores the feature using the decoder. We design an encoder and a decoder as a $3 \times 3$ convolution layer to compress the channel size depending on the compression ratio $N$. The compressed feature $\mathbf{F}'_l \in \mathbb{R}^{H \times W \times (C/N)}$ and decompressed feature $\bar{\mathbf{F}}_l \in \mathbb{R}^{H \times W \times C}$ are determined by

$$\mathbf{F}'_l = f_{\text{encoder}}(\mathbf{F}_l), \quad l = 1, ..., L, \qquad \bar{\mathbf{F}}_l = f_{\text{decoder}}(\mathbf{F}'_l), \quad l = 1, ..., L.$$

**Transformation.** The ego agent transforms the decoded features into the ego agent coordinate. Also, features outside the detection range are cropped to delete unnecessary information. After this process, we use a spatial-temporal correction module (STCM) used in Xu et al. [2022b] to compensate for the time delay caused by communication latency. We denote by $\Gamma(\cdot)$ the sequential process of transform, crop, and STCM. By concatenating $\Gamma(\bar{\mathbf{F}}_l)$ along the agent dimension, the multi-agent feature $\mathbf{M}^0 \in \mathbb{R}^{L \times H \times W \times C}$ is generated. That is,

$$\mathbf{M}^0 = \underset{l \in [1, L]}{\|} \Gamma(\bar{\mathbf{F}}_l),$$

where $\|_l (\cdot)$ denote the concatenation along the agent dimension.

### 3.4 Fusion Module

The overall architecture of the fusion module is described in Figure 3 (b). Our fusion module in ParCon mainly consists of three sub-modules: agent-wise attention (A-Att), spatial-wise attention (S-Att), and spatial-wise convolution layer (S-Conv). Each sub-module generates inter-agent complementary information, spatial global context, and spatially detailed information, respectively. We refer to the output of each sub-module as the *intent* of each sub-module. To prevent the model from being too heavy due to two transformer sub-modules (A-Att, S-Att), we use a channel-wise compression layer (CCL) to generate a compressed feature which is compressed the input feature channel-wise. This is one of the key components that makes our model efficient while maintaining state-of-the-art performance. After intents are generated from sub-modules, the output feature concatenates all intents and the compressed feature.

**Channel Compression Layer (CCL).** To reduce the input feature size in a simple and information-maintained manner, we design CCL as one fully connected (FC) layer that compresses the feature channel-wise. The input

to the fusion module is the multi-agent feature $\mathbf{M}^0$, and the input at the $d$-th depth is $\mathbf{M}^d \in \mathbb{R}^{L \times H \times W \times C}$, $d = 1, ..., D-1$ where $D$ is the maximum depth of the fusion module. Then, the compressed feature $\mathbf{M}'$, output of CCL, at the $d$-th depth for each $d = 0, ..., D-1$ is given by

$$\mathbf{M}' = f_{\mathrm{CCL}}(\mathbf{M}^d) \in \mathbb{R}^{L \times H \times W \times C/4}.$$

Although CCL is a simple FC layer, it plays two important roles in ParCon. First, by compressing the feature channel size from $C$ to $C/4$, CCL significantly reduces the number of parameters and GFLOPs in ParCon. Second, because CCL mixes various intents to generate $\mathbf{M}'$, it contributes to improved performance by compensating for the deficiencies of individual modules and enhancing overall robustness.

**Heterogeneous Relative Pose Encoding (HRPE).** Several factors, such as 1) agent type, 2) relative angle, and 3) distance, affect the point-cloud distribution and, therefore, change the feature distribution. HRPE offers these three pieces of information to consider the inter-agent relationship.

According to the $l$-th agent type $t_l \in \{\mathrm{I}, \mathrm{V}\}$, we define the period $w_j$ as the inverse of the constant hyperparameter $\tau \in \mathbb{R}$ to some powers of integer $j \in [0, C/16 - 1]$. In particular, $\omega_j = 1/(\tau^{2j+1})$ if the agent type of $l$ is infrastructure, and $\omega_j = 1/(\tau^{2j})$ if the agent type of $l$ is vehicle. Using the approach in Piergiovanni et al. [2023], we make fixed encoding values $\mathbf{p} \in \mathbb{R}^{C/4}$ of the relative angle $\theta$ and distance $d$ between the ego and aux agents as

$$\mathbf{p}[4j : 4(j+1)] = [\sin(d * \omega_j), \cos(d * \omega_j), \sin(\theta * \omega_j), \cos(\theta * \omega_j)],$$

where $*$ is the multiplication. The fixed encoding value $\mathbf{p}$ is added to the $l$-th compressed input feature $\mathbf{M}'_l \in \mathbb{R}^{H \times W \times C/4}$, and then, the encoded feature $\mathbf{M}^{\mathrm{pos}} \in \mathbb{R}^{L \times H \times W \times C/4}$ concatenates the added feature along the agent dimension. That is, $\mathbf{M}^{\mathrm{pos}} = \underset{l \in [1, L]}{\|} (\mathbf{M}'_l + \mathbf{p})$.

**Agent-wise Attention Module (A-Att).** As shown in Figure 3 (a), we use the vanilla attention module available in the code of Xu et al. [2022b], which we denote by $\mathrm{CAV\_ATT}^d$ at the $d$-th depth. This module does not distinguish agent type and utilizes a multi-head self-attention transformer. The output $\mathbf{M}^{\mathbf{AA}}$ of A-Att at the $d$-th depth is given by

$$\mathbf{M}^{\mathbf{AA}} = \begin{cases} \mathrm{CAV\_ATT}^1(\mathbf{M}^{\mathrm{pos}}) + \mathbf{M}', & d = 1 \\ \mathrm{CAV\_ATT}^d(\mathbf{M}') + \mathbf{M}', & d \neq 1. \end{cases}$$

We use HRPE in the first depth of the fusion module to better interpret the other agent features.

**Spatial-wise Attention Module (S-Att).** We use Dilated Neighborhood Attention Transformer (DiNAT) (Hassani & Shi [2022]) to interpret a global context. DiNAT maintains translation equivariance and reduces computation burden by resembling the convolution operation. Also, it efficiently widens the receptive fields by using the dilation rate. As shown in Figure 3 (b), we use one block of the original DiNAT module and make it lighter by reducing the depth, kernel size, and dilation value of the original DiNAT to enhance the overall efficiency of the model. The output $\mathbf{M}^{\mathbf{SA}}$ of S-Att is determined by

$$\mathbf{M}^{\mathbf{SA}} = \mathrm{DiNAT}(\mathbf{M}') + \mathbf{M}'.$$

**Spatial-wise Convolution Module (S-Conv).** We use convolution layers to capture the spatial local information. As shown in Figure 3 (c), S-Conv uses three blocks of convolution layers with batch normalization and activate function ReLU. Also, we apply residual connections only to the first and second blocks while omitting them in the third block. The output $\mathbf{M}^{\mathbf{SC}}$ of S-Conv is determined by

$$\mathbf{M}^{\mathbf{SC}} = \mathrm{S\_Conv}(\mathbf{M}')$$

**Parallel Connection.** ParCon concatenates the intents of the three sub-modules and the compressed original feature $\mathbf{M}'$. Then, the output of $d$-th depth fusion module $\mathbf{M}^d$ is generated by applying a multilayer perceptron (MLP) with the residual connection. That is,

$$\mathbf{M}^{\mathbf{O}} = \mathbf{M}^{\mathbf{AA}} \parallel \mathbf{M}^{\mathbf{SA}} \parallel \mathbf{M}^{\mathbf{SC}} \parallel \mathbf{M}', \qquad \mathbf{M}^d = \mathrm{MLP}(\mathbf{M}^{\mathbf{O}}) + \mathbf{M}^{\mathbf{O}}, \quad d = 1, ..., D.$$

## 3.5 Head

After passing the last depth, we get the final output $\mathbf{M}^D$. We extract the ego agent output $\mathbf{M}_1^D$ and apply two $1 \times 1$ convolution layers, $f_{\mathrm{head}}^{\mathrm{cls}}(\cdot)$ and $f_{\mathrm{head}}^{\mathrm{reg}}(\cdot)$, for the detection box classification and regression, respectively. Specifically, $f_{\mathrm{head}}^{\mathrm{cls}}(\cdot)$ makes the classification output tensor $\hat{\mathbf{Y}}_{\mathrm{cls}} \in \mathbb{R}^{H \times W \times 2}$ that identifies whether an object is a vehicle or not, and $f_{\mathrm{head}}^{\mathrm{reg}}(\cdot)$ makes the regression output tensor $\hat{\mathbf{Y}}_{\mathrm{reg}} \in \mathbb{R}^{H \times W \times 14}$ that contains center of boxes $(x, y, z)$, size of boxes $(w, l, h)$ and heading of vehicle $\phi$. We use the total loss $\mathcal{L}_{\mathrm{total}} = \mathcal{L}_{\mathrm{cls}} + \mathcal{L}_{\mathrm{reg}}$ where $\mathcal{L}_{\mathrm{cls}}$ is calculated using $\hat{\mathbf{Y}}_{\mathrm{cls}}$ and focal loss (Lin et al. [2017]) and $\mathcal{L}_{\mathrm{reg}}$ is calculated using $\hat{\mathbf{Y}}_{\mathrm{reg}}$ and the smooth L1 loss (Berrada et al. [2018]).

## 4 Experiments

### 4.1 Experimental Settings

**Comparison Models and Datasets.** We compare our proposed architecture with state-of-the-art (SOTA) models, CoBEVT Xu et al. [2022a], V2X-ViT Xu et al. [2022b], Where2comm Hu et al. [2022], CoAlign Lu et al. [2023], on the datasets of V2XSet (Xu et al. [2022b]), OPV2V (Xu et al. [2022c]), and DAIR-V2X (Yu et al. [2022]). V2XSet is a simulated dataset, including V2X scenarios, using the CARLA simulator (Dosovitskiy et al. [2017]) and OpenCDA (Xu et al. [2021]). OPV2V is a simulated dataset, only including V2V scenarios. Also, DAIR-V2X is a real-world dataset, including V2X scenarios, collected from one actual vehicle and one road infrastructure. We set the LiDAR detection range as $x \in [-140.8, 140.8]$ and $y \in [-38.4, 38.4]$ on V2XSet and OPV2V, and $x \in [-102.4, 102.4]$ and $y \in [-38.4, 38.4]$ on DAIR-V2X. A detailed description of the datasets and models is in Appendix B.1 and B.2.

**Evaluation Metrics and Noise Settings.** We calculate the Average Precision (AP) at the Intersection-over-Union (IoU) thresholds of 0.5 and 0.7 to evaluate 3D detection accuracy and use the number of parameters (# Params) and GFLOPs to compare model efficiency. To consider communication noise while sharing the feature of connected agents, we add communication noise to training and inference data, which includes latency, heading noises, and localization noises. The latency is based on the total time delay used in Xu et al. [2022b] and is set to follow uniform distribution $U(0, t_{\mathrm{lag.}})$ with the maximum latency $t_{\mathrm{lag.}}$. The heading and localization noises follow normal distributions with zero mean and standard deviations $\sigma_{\mathrm{hdg.}}$ and $\sigma_{\mathrm{loc.}}$, that is, $\mathcal{N}(0, \sigma_{\mathrm{hdg.}}^2)$ and $\mathcal{N}(0, \sigma_{\mathrm{loc.}}^2)$, respectively. We use three different noise settings: perfect, mild noise, and harsh noise. In the perfect setting, we let $t_{\mathrm{lag.}} = 0, \sigma_{\mathrm{hdg.}} = 0, \sigma_{\mathrm{loc.}} = 0$, and in the mild noise setting, we let $t_{\mathrm{lag.}} = 200, \sigma_{\mathrm{hdg.}} = 0.2, \sigma_{\mathrm{loc.}} = 0.2$. In the harsh noise setting, we use different values of $t_{\mathrm{lag.}} \in [0, 500], \sigma_{\mathrm{hdg.}} = [0, 1.0], \sigma_{\mathrm{loc.}} = [0, 0.5]$. Also, we train three types of models based on the noise settings: perfect, noise, and fine-tuned. The perfect model is trained in the perfect setting, the noise model in the simple noise setting (see Appendix B.3 for details), and the fine-tuned model fine-tunes the perfect model with the mild noise setting. We validate the perfect model, noise model, and fine-tuned model with the perfect setting, mild noise setting, and harsh noise setting, respectively.

**Training Parameters.** For a fair comparison, we train both our models and SOTA models using AdamW (Loshchilov & Hutter [2017]), with a learning rate of 3e-4 and a weight decay of 0.01. We use the same learning scheduler, Cosine Annealing Warm-Up Restarts (Loshchilov & Hutter [2016]), applying a warm-up learning rate of 2e-4. Regarding V2XSet and OPV2V, we train the models for up to 40 epochs and 10 warm-up epochs. We also train the models for up to 20 epochs and 5 warm-up epochs for DAIR-V2X. The models are trained and validated on an RTX 4090.

Table 1: Comparison of detection accuracy on V2XSet, OPV2V, and DAIR-V2X.

| Model | Perfect | | | Mild Noise | | |
|---|---|---|---|---|---|---|
| | V2XSet AP@0.5/0.7 | OPV2V AP@0.5/0.7 | DAIR-V2X AP@0.5/0.7 | V2XSet AP@0.5/0.7 | OPV2V AP@0.5/0.7 | DAIR-V2X AP@0.5/0.7 |
| No Fusion | 0.754 / 0.602 | 0.659 / 0.545 | 0.562 / 0.452 | 0.754 / 0.602 | 0.659 / 0.545 | 0.562 / **0.452** |
| Late Fusion | 0.885 / 0.764 | 0.852 / 0.768 | 0.573 / 0.397 | 0.844 / 0.552 | 0.726 / 0.539 | 0.549 / 0.364 |
| CoBEVT | 0.880 / 0.825 | 0.852 / 0.787 | 0.570 / 0.452 | 0.787 / 0.600 | 0.698 / 0.522 | 0.550 / 0.407 |
| V2X-ViT | 0.866 / 0.773 | 0.843 / 0.763 | 0.603 / 0.455 | 0.811 / 0.648 | 0.755 / 0.615 | 0.572 / 0.409 |
| Where2comm | **0.904** / 0.835 | 0.858 / 0.789 | 0.594 / 0.429 | **0.852** / 0.668 | 0.760 / 0.576 | 0.562 / 0.384 |
| CoAlign | 0.888 / 0.799 | 0.819 / 0.741 | 0.578 / 0.433 | 0.816 / 0.605 | 0.726 / 0.539 | 0.557 / 0.397 |
| ParCon (Ours) | 0.891 / **0.839** | **0.888 / 0.830** | **0.613 / 0.482** | 0.850 / **0.707** | **0.790 / 0.649** | **0.596** / 0.438 |

Table 2: Comparison of efficiency.

| Model | # Params | GFLOPs |
|---|---|---|
| CoBEVT | 11.14M | 213 |
| V2X-ViT | 14.17M | 287 |
| Where2comm | 8.69M | 177 |
| CoAlign | 12.94M | 96 |
| ParCon(Ours) | 5.57M | 85 |

Table 3: Performance comparison between ParCon and ParCon-S at AP@0.7. ParCon-S is the sequential architecture model using the same sub-modules of ParCon.

| Model | V2XSet | OPV2V | DAIR-V2X |
|---|---|---|---|
| ParCon | **0.707** | **0.649** | **0.438** |
| ParCon-S | 0.690 | 0.642 | 0.430 |

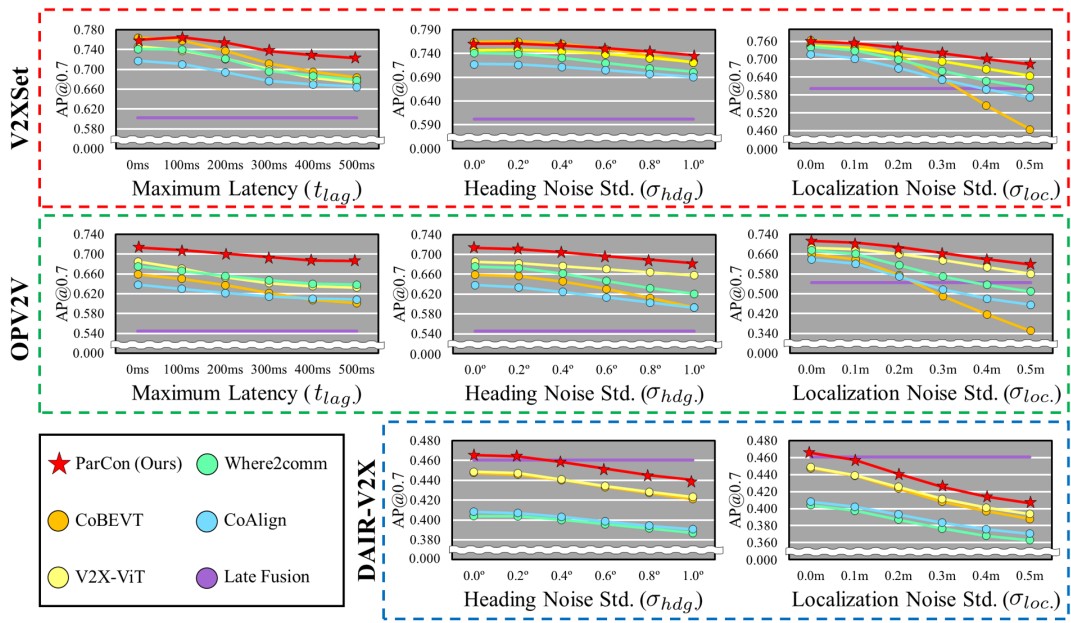

Figure 4: Robustness in various noise ranges on V2XSet, OPV2V, and DAIR-V2X.

**Model Parameters.** After the feature extraction, we compress the channel size of the feature from $C = 256$ to $C/N = 8$ for the feature sharing. We set the channel size of multi-agent feature $\mathbf{M^0}$ as 256. Regarding the fusion module, HRPE in A-att utilizes the relative angle $\theta$ and distance $d$. For V2XSet and OPV2V, we divide $\theta$ and $d$ into $20°$ and $25$ m, and into $10°$ and $15$ m for DAIR-V2X. In S-Att, the lightweight DiNAT (Hassani & Shi [2022]) features two depths. For DiNAT's hyperparameters, we use a $7 \times 7$ kernel with dilation rates of 4 and 2 applied to each depth, respectively. The S-Conv comprises three blocks, each consisting of a convolution layer with a $3 \times 3$ kernel.

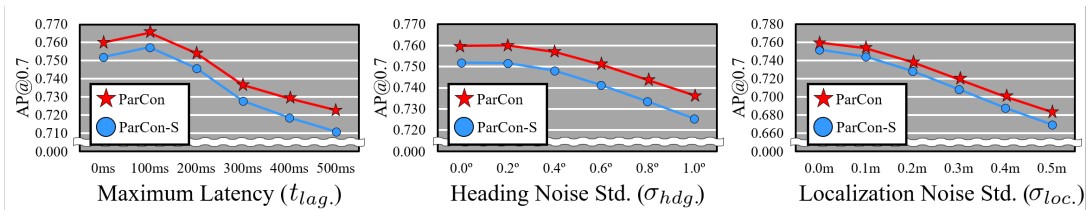

Figure 5: Robustness comparison between ParCon and ParCon-S in various noise ranges on V2XSet. ParCon-S is the sequential architecture model using the same sub-modules of ParCon.

## 4.2 QUANTITATIVE EVALUATION

**Detection Accuracy.** Table 1 compares the proposed ParCon with the other SOTA methods on the simulation datasets, V2XSet and OPV2V, and the real-world dataset, DAIR-V2X. We also consider single-agent detection (No Fusion) and Late Fusion. We observe the following. i) In the simulated datasets, in the mild noise setting, ParCon achieves the SOTA accuracy increased by 5.83% compared with Where2comm on V2XSet and by 5.52% compared with V2X-ViT on OPV2V at AP@0.7. ii) Even in the real-world dataset, in the mild noise setting, ParCon achieves the SOTA accuracy increased by 6.91% compared with V2X-ViT at AP@0.7. In summary, our proposed collaborative perception model, ParCon, outperforms the SOTA models on all the datasets. iii) Our model achieves the highest performance at AP@0.7, except in the mild noise setting of the DAIR-V2X dataset. In this specific case, the No Fusion approach shows the highest performance. This is because the DAIR-V2X dataset includes only one vehicle and one infrastructure, providing limited opportunity for improvement through collaboration. Thus, all the fusion methods, including ours, do not outperform the No Fusion approach in this scenario.

**Model Efficiency.** Table 2 compares model efficiency in terms of # Params and GFLOPs. The # Params in ParCon is reduced by 56.95% compared with CoAlign and by 60.69% compared with V2X-ViT. The GFLOPs of ParCon is reduced by 11.46% compared with CoAlign and 70.38% compared with V2X-ViT. The efficiency stems from our CCL, which compress the channel size from 256 to 64.

**Noise Robustness.** In Figure 4, we compare the noise robustness on the V2XSet, OPV2V, and DAIR-V2X datasets. ParCon always outperforms the SOTA models across all the datasets, even in the real-world dataset DAIR-V2X. At the same time, ParCon shows strong noise robustness (the lowest accuracy drop) among the SOTA models on the simulated dataset. From zero noise to the maximum noise value, the accuracy (AP@0.7) of ParCon decreases by 4.90%/3.08%/10.01% on V2XSet and by 3.78%/4.32%/13.41% on OPV2V under latency/heading noise/localization noise, respectively. In contrast, V2X-ViT experiences larger drops of 9.78%/3.38%/13.62% on V2XSet and 7.07%/3.95%/15.18% on OPV2V. Similarly, Where2comm shows even greater declines, with decreases of 8.42%/5.37%/18.42% on V2XSet and 5.48%/8.12%/24.55% on OPV2V. You can see detailed values in Appendix C.2.

**Importance of Parallel Connection.** To identify the effectiveness of parallel connection, we design the sequential architecture model, ParCon-S, which uses the same sub-modules of ParCon and connects them in series. Similar to ParCon, ParCon-S compresses the multi-agent feature with CCL and uses a fully connected layer to restore the channel size of the output feature to match the channel size of the original feature. Table 3 and Figure 5 show the comparison in terms of detection accuracy and noise robustness between ParCon and ParCon-S, respectively. As shown in Table 3, the parallel model always outperforms the sequential model at AP@0.7. The parallel model enhances the performance by 2.38%, 1.17%, and 1.82% compared to the sequential model on the V2XSet, OPV2V, and DAIR-V2X datasets, respectively. Furthermore, as shown in Figure 5, the parallel model outperforms the sequential model across various noise levels. In the harsh noise setting, the accuracy of ParCon drops by 4.91%/3.08%/10.08% under latency/heading noise/localization noise, respectively, while ParCon-S experiences larger drops of 5.46%/3.51%/11.01% (See Appendix C.2 for detail values). This comparison demonstrates that the parallel connection consistently outperforms the sequential connection across all datasets and noise settings.

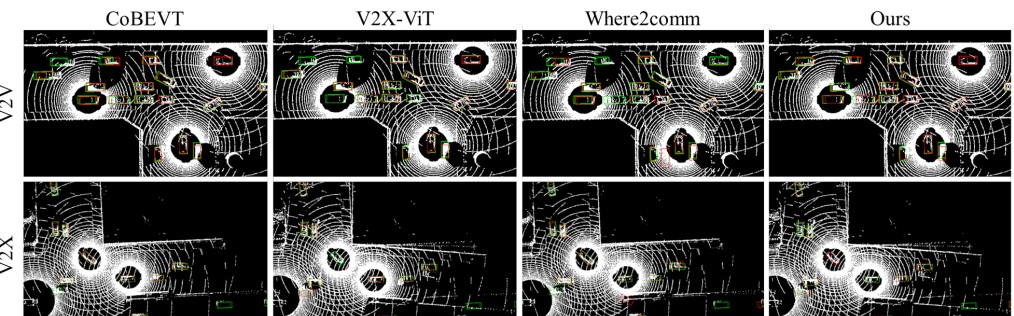

Figure 7: **Visualization of detected boxes in the V2XSet Dataset.** The green boxes represent the ground truth, while the red ones represent prediction. Our model ParCon shows more precise detection.

### 4.3 QUALITATIVE EVALUATION

**Investigation of CCL Weights.** At CCL, the concatenated intents ($\mathbf{M^{AA}} \parallel \mathbf{M^{SA}} \parallel \mathbf{M^{SC}} \parallel \mathbf{M'}$) are fused each other channel-wise. As we visualize the weights of CCL in Figure 6, it is noteworthy that after the second depth, the weights appear divided into four sections corresponding to the concatenated intents. Also, we observe that i) as shown in Table 4 and Table 5, the summation of the absolute weights in the third section of CCL, corresponding to the intent of S-Conv, shows the highest score, and ii) the summation of the absolute weights in the third section of CCL in the fine-tuned model is higher than in the perfect model. These observations suggest that S-Conv, which generates spatial local information, plays a crucial role in noise rejection, and that CCL leverages this by focusing more on the intent of S-Conv. In essence, CCL aggregates the intents of specific effective modules to enhance overall performance and compensate for the limitations of other modules.

**Visualization of Detection Results.** Figure 7 shows the detection visualization of CoBEVT, V2V-ViT, and Where2comm in V2V and V2X scenarios of V2XSet. Our model accurately predicts bounding boxes that are well-aligned with the ground truth and successfully detects objects under occlusion. In contrast, other models display misalignments and fail to detect objects when occlusion occurs.

### 4.4 ABLATION STUDIES

**CCL compression ratio.** We train ParCon with various compression rates in CCL to validate whether the reduction in feature size causes information loss. Table 6 shows that ParCon with the rate of $\times 4$ yields the best

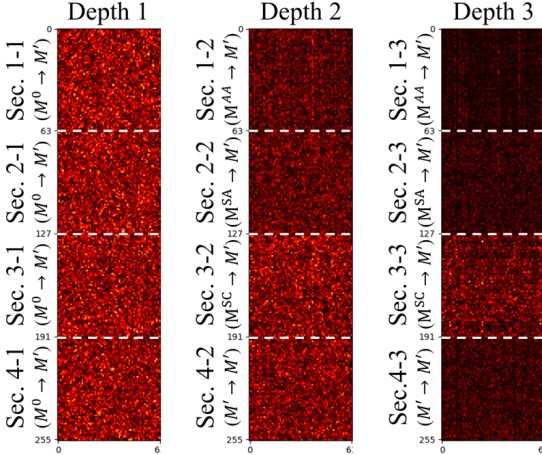

Figure 6: Visualization of the weights of CCL.

Table 4: Summation of the absolute value of CCL weights in the perfect setting.

| Section | Depth 1 | Depth 2 | Depth 3 |
|---------|---------|---------|---------|
| Sec.1 | 190.153 | 104.393 | 59.076 |
| Sec.2 | 190.497 | 117.089 | 71.186 |
| Sec.3 | 191.221 | 142.863 | 121.540 |
| Sec.4 | 189.761 | 138.970 | 81.123 |

Table 5: Summation of the absolute value of CCL weights under the noise setting.

| Section | Depth 1 | Depth 2 | Depth 3 |
|---------|---------|---------|---------|
| Sec.1 | 228.254 | 103.984 | 48.986 |
| Sec.2 | 226.726 | 117.295 | 70.545 |
| Sec.3 | 224.471 | 172.468 | 145.967 |
| Sec.4 | 221.155 | 158.333 | 84.261 |

Table 6: Effect of compression ratio at AP@0.7.

| Rate | Perfect | Mild Noise | # Params | GFLOPs |
|---|---|---|---|---|
| ×1 | 0.839 | 0.670 | 19.69M | 451 |
| ×2 | 0.838 | 0.687 | 8.74M | 161 |
| ×4 (Ours) | 0.850 | **0.707** | 5.57M | 85 |
| ×8 | **0.851** | 0.689 | 5.00M | 64 |

Table 7: Effect of HRPE in the mild noise setting on V2XSet.

| HRPE | AP@0.5 | AP@0.7 |
|---|---|---|
| ✓ | 0.850 | 0.707 |
|  | 0.844 | 0.694 |

Table 8: Effect of sub-modules in the mild noise setting.

| A-Att | S-Att | S-Conv | Original | AP@0.5 | AP@0.7 |
|---|---|---|---|---|---|
| ✓ | ✓ | ✓ | ✓ | **0.850** | **0.707** |
|  | ✓ | ✓ | ✓ | 0.774 | 0.645 |
| ✓ |  | ✓ | ✓ | 0.848 | 0.702 |
| ✓ | ✓ |  | ✓ | 0.813 | 0.663 |
| ✓ | ✓ | ✓ |  | 0.842 | 0.683 |

Table 9: Ablation study of S-Att. Whether to apply the S-Att or vanilla model (ViT) in the mild noise setting on V2XSet.

| S-Att | AP@0.5 | AP@0.7 | # Params | GFLOPs |
|---|---|---|---|---|
| ✓ | 0.850 | 0.707 | 5.57M | 85 |
|  | 0.809 | 0.660 | 11.79M | 75 |

performance at AP@0.7. Compared with Parcon with the rate of ×1, Parcon with the rate of ×4 yields the accuracy enhanced by 5.50% in the mild noise setting, and the number of parameters and GFLOPs decreased by 71.68% and 81.15%, respectively. Based on this result, we select the rate of ×4 as the final setting.

**Effect of HRPE.** We use HRPE in A-Att to help the module understand the features of the other agents. Table 7 shows that HRPE enhances the performance of the ParCon in the mild noise setting. We notice that HRPE does not contribute much in the perfect setting. By appropriately setting the two hyperparameters for dividing $\theta$ and $d$, HRPE makes our model robust to noise.

**Effect of sub-modules.** We present the ablation study results in Table 8. The inter-agent complementary information, which is related to A-Att, affects the performance of ParCon significantly. The performance AP@0.7 decreases by 8.76% by eliminating A-Att. The next contributing information is due to S-Conv; if S-Conv is absent, the performance AP@0.7 decreases by 6.12%.

**Effect of S-Att.** In Table 9, we compare our S-Att module with ViT (Dosovitskiy et al. [2020]), which is vanilla model of S-Att. Although using S-Att reduces the number of parameters by 52.76%, it enhances the performance of AP@0.7 by 7.10% in the mild noise setting.

## 5 CONCLUSION

This paper has proposed a new collaborative perception model, ParCon, featuring the parallel connection of sub-modules. With the parallel architecture, ParCon effectively handles communication noises, such as latency, heading noise, and localization noise. Also, ParCon is efficient in terms of the number of parameters and GFLOPs as it uses CCL and S-Att. CCL reduces the feature size to make our fusion module highly lightweight with enhanced accuracy under the noise. We adopt DiNAT (Hassani & Shi [2022]) to capture the spatial global context in S-Att, which enhances the performance while simultaneously reducing the number of parameters compared to the vanilla model, ViT (Dosovitskiy et al. [2020]). Our work provides an important guideline for designing new collaborative perception architectures by demonstrating the benefits of parallel architecture, CCL, and S-Att, as well as their effectiveness and efficiency in handling communication noise.

**Limitation and future work.** To focus on the effectiveness of the parallel architecture, we use the same sub-modules as in other SOTA models, with the exception of S-Att. In future work, we plan to design advanced sub-modules that can enhance synergy with the parallel architecture.

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

## A  APPENDIX

## B  IMPLEMENTATION DETAILS

### B.1  DATASET.

We use the V2XSet Xu et al. [2022b], OPV2V Xu et al. [2022c], and DAIR-V2X Yu et al. [2022] datasets to train and validate models in both V2V and V2X scenarios.

**V2XSet.** V2XSet is a simulated dataset supporting V2X perception, co-simulated using CARLA Dosovitskiy et al. [2017] and OpenCDA Xu et al. [2021]. It comprises 73 scenes with a minimum of 2 to 5 connected agents and includes 11K 3D annotated LiDAR point cloud frames. The training, validation, and testing sets consist of 6.7K, 2K, and 2.8K frames, respectively.

**OPV2V.** OPV2V designed for multi-agent V2V perception. Each frame typically comprises approximately 3 CAVs, with a minimum of 2 and a maximum of 7. It includes 10.9K LiDAR point cloud frames with 3D annotations. The training, validation, and testing splits include 6.8K, 2K, and 2.2K frames, respectively.

**DAIR-V2X.** DAIR-V2X is a real-world dataset for collaborative perception. The dataset includes 9K frames from a vehicle and a road infrastructure, which is equipped with both a LiDAR and a 1920x1080 camera. The LiDAR of infrastructure is 300-channel, and the vehicle's is 40-channel.

## B.2 COMPARISON MODELS.

We select CoBEVT Xu et al. [2022a], V2X-ViT Xu et al. [2022b], Where2comm Hu et al. [2022], CoAlign Lu et al. [2023] to compare ParCon.

**CoBEVT.** CoBEVT presents a framework for multi-agent multi-camera perception that collaboratively produces BEV map predictions using both camera and LiDAR.

**V2X-ViT.** V2X-viT attempts to utilize heterogeneous agents, such as vehicles and road infrastructure, and solve the problem of heterogeneity between them using a transformer with a graph structure.

**Where2comm.** Where2comm introduces a spatial confidence map, enabling agents to share only spatially sparse yet perceptually important information.

**CoAlign.** CoAlign introduces a hybrid framework combining intermediate and late fusion, and leverages an agent-object pose graph to align and optimize relative poses between agents, making the system more robust to noise introduced by pose estimation errors.

## B.3 NOISE SETTINGS AND TRAINED MODELS.

We define four different types of noise as shown in Table 10, and three different types of trained model as shown in Table 11. The latency is based on the total time delay used in Xu et al. [2022b] and is set to follow uniform distribution $U(0, t_{\text{lag.}})$ with the maximum latency $t_{\text{lag.}}$. The heading and localization noises follow normal distributions with zero mean and standard deviations $\sigma_{\text{hdg.}}$ and $\sigma_{\text{loc.}}$, that is, $\mathcal{N}(0, \sigma_{\text{hdg.}}^2)$ and $\mathcal{N}(0, \sigma_{\text{loc.}}^2)$, respectively. We use three different noise settings: Perfect, Mild Noise, and Harsh Noise. In the perfect setting, we let $t_{\text{lag.}} = 0, \sigma_{\text{hdg.}} = 0, \sigma_{\text{loc.}} = 0$, and in the mild noise setting, we let $t_{\text{lag.}} = 200, \sigma_{\text{hdg.}} = 0.2, \sigma_{\text{loc.}} = 0.2$. In the harsh noise setting, we use different values of $t_{\text{lag.}} \in [0, 500], \sigma_{\text{hdg.}} = [0, 1.0], \sigma_{\text{loc.}} = [0, 0.5]$. Also, we train the three types of models based on the noise setting used in the training: Perfect, Noisy, and Fine-tuned. The perfect model is trained with the perfect setting, the noisy model with the simple noise setting, and the fine-tuned model is fine-tuning the perfect model with the mild noise setting.

Table 10: Various types of noise setting.

| Noise Setting | Latency | Hdg. Noise | Loc. Noise |
|---|---|---|---|
| Perfect | 0 | 0 | 0 |
| Simple Noise | 100 | $\mathcal{N}(0, 0.2^2)$ | $\mathcal{N}(0, 0.2^2)$ |
| Mild Noise | $U(0, 200)$ | $\mathcal{N}(0, 0.2^2)$ | $\mathcal{N}(0, 0.2^2)$ |
| Harsh Noise | $U(0, t_{\text{lag.}})$ $t_{\text{lag.}} \in [0, 500]$ | $\mathcal{N}(0, \sigma_{\text{hdg.}}^2)$ $\sigma_{\text{hdg.}} \in [0.0, 1.0]$ | $\mathcal{N}(0, \sigma_{\text{loc.}}^2)$ $\sigma_{\text{loc.}} \in [0.0, 0.5]$ |

Table 11: Various types of trained model.

| Training Type | Method |
|---|---|
| Perfect | Training model in the perfect setting. |
| Noise | Training model in the simple noise setting. |
| Fine-tuned | First, training model in the perfect setting. Then, fine-tuning model in the mild noise setting. |

## C  EXPERIMENTS.

### C.1  ATTENTION WEIGHTS VISUALIZATION.

In Figure 8, we visualize the attention weights in A-Att, where the brighter colors indicate the areas that the ego agent focuses on. Some observations to notice are as follows: i) Our models show a clear difference between the regions that are paid attention to or not, compared to the V2X-ViT's attention weights (Figure 8-(b)). ii) Our models have different attention tendencies. For ParCon in Figure 8-(d), the ego agent relies on its own confidence regions where its point clouds are dense and not occluded. Also, it relies on the aux agent's confidence regions to incorporate wide-range information. In the sequential model in Figure 8-(c), the ego agent focuses on the areas where objects densely exist. The tendencies are more explicitly identified in the infrastructure data, which has a wider detection range but is sparse and can be noisy at far distances. In ParCon, the ego agent takes the infra data relatively less, while the ego agent in the sequential model utilizes the infrastructure data heavily because the infra data has lower confidence than the data from nearby aux agents.

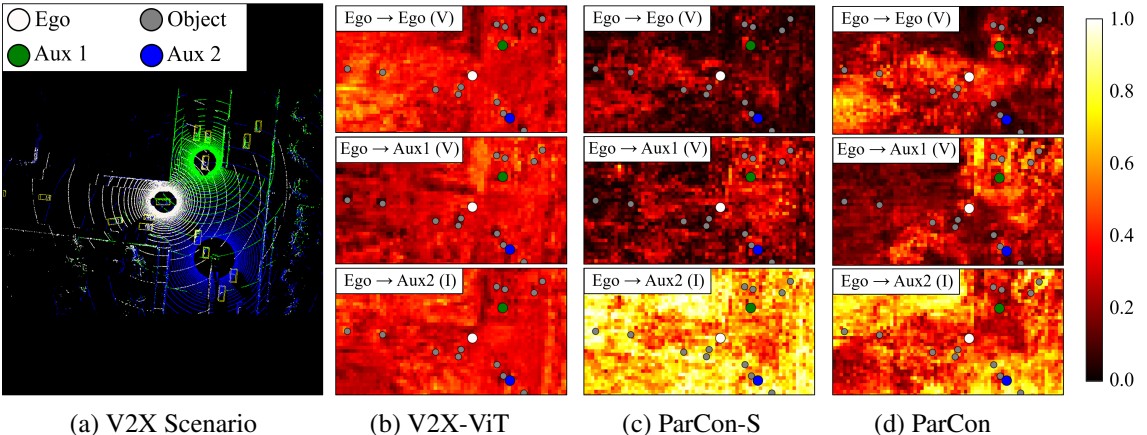

|            |              |             |            |
|:----------:|:------------:|:-----------:|:----------:|
| (a) V2X Scenario | (b) V2X-ViT | (c) ParCon-S | (d) ParCon |

Figure 8: **Aggregated LiDAR points and attention map.** The white circle is the ego agent, and the blue and green circles are aux agents. Small grey circles are vehicles in the detection range.

### C.2  NOISE ROBUSTNESS.

As shown in Table 12, 13, 14, and 18, we show the detailed values used in Section 4.2, which is ParCon outperforms the SOTA models across all noise ranges.

To support our claims of noise robustness, we analyze an additional comparison, and then divide the noise set into two subsets: 1) normal noise and 2) severe noise. The normal noise subset is based on the normal range defined in the work Xu et al. [2022b]. Hence, we define a normal noise subset as $t_{\text{lag.}} \in [100\,\text{ms}, 200\,\text{ms}]$, $\sigma_{\text{loc.}} \in [0.1\,\text{m}, 0.2\,\text{m}]$, and $\sigma_{\text{hdg.}} \in [0.2°, 0.4°]$, and the severe noise subset as $t_{\text{lag.}} \in [300\,\text{ms}, 500\,\text{ms}]$, $\sigma_{\text{loc.}} \in [0.3\,\text{m}, 0.5\,\text{m}]$, and $\sigma_{\text{hdg.}} \in [0.6°, 1.0°]$. In Table 15, 16, 17, and 19, we evaluate performance by averaging each component in the subset.

**Normal Noise.** Under normal noise conditions, ParCon outperforms SOTA models in terms of latency, heading, and localization noise. Moreover, ParCon demonstrates the lowest sensitivity to noise, decreasing by 0.03%, 0.18%, and 1.81& on V2XSet, 1.23%, 0.77%, and 2.28% on OPV2V under latency/heading noise/localization noise, and 0.92%, 3.49% on DAIR-V2X under heading/localization noise, repectively.

Our models exhibit performance enhancements regarding localization and heading noise, while most SOTA models show reduced performance.

**Severe Noise.** Our models always outperform the SOTA models in severe noise conditions. Also, our models show the lowest sensitivity to severe noise. ParCon has low drop rate by 4.01%/2.13%/7.75% on V2XSet and by 3.38%/3.32%/10.29% on OPV2V under latency/heading noise/localization noise, respectively. Although the drop rate of our method is not always the lowest in some dataset, ParCon always keeps SOTA performance.

Table 12: Robustness to various ranges of noises with detailed values on V2XSet.

| Model | Maximum Latency ($t_{\text{lag.}}$) | | | | | |
|---|---|---|---|---|---|---|
| | 0 ms | 100 ms | 200 ms | 300 ms | 400 ms | 500 ms |
| CoBEVT | **0.764** | 0.759 | 0.738 | 0.712 | 0.696 | 0.684 |
| V2X-ViT | 0.747 | 0.738 | 0.721 | 0.701 | 0.683 | 0.673 |
| Where2comm | 0.741 | 0.741 | 0.722 | 0.697 | 0.688 | 0.679 |
| CoAlign | 0.718 | 0.711 | 0.695 | 0.677 | 0.670 | 0.665 |
| ParCon (Ours) | 0.760 | **0.765** | **0.754** | **0.737** | **0.729** | **0.723** |

| Model | Heading Noise std. ($\sigma_{\text{loc.}}$) | | | | | |
|---|---|---|---|---|---|---|
| | $0.0°$ | $0.2°$ | $0.4°$ | $0.6°$ | $0.8°$ | $1.0°$ |
| CoBEVT | **0.764** | **0.766** | **0.760** | 0.750 | 0.735 | 0.721 |
| V2X-ViT | 0.747 | 0.748 | 0.745 | 0.739 | 0.731 | 0.721 |
| Where2comm | 0.741 | 0.740 | 0.731 | 0.720 | 0.710 | 0.701 |
| CoAlign | 0.718 | 0.717 | 0.712 | 0.705 | 0.698 | 0.691 |
| ParCon (Ours) | 0.760 | 0.760 | 0.757 | **0.751** | **0.744** | **0.737** |

| Model | Localization Error std. ($\sigma_{\text{hdg.}}$) | | | | | |
|---|---|---|---|---|---|---|
| | 0.0 m | 0.1 m | 0.2 m | 0.3 m | 0.4 m | 0.5 m |
| CoBEVT | **0.764** | 0.752 | 0.711 | 0.634 | 0.545 | 0.465 |
| V2X-ViT | 0.747 | 0.737 | 0.717 | 0.693 | 0.668 | 0.645 |
| Where2comm | 0.741 | 0.730 | 0.700 | 0.661 | 0.628 | 0.605 |
| CoAlign | 0.718 | 0.703 | 0.671 | 0.632 | 0.599 | 0.573 |
| ParCon (Ours) | 0.760 | **0.754** | **0.738** | **0.719** | **0.700** | **0.683** |

Table 13: Robustness to various ranges of noises with detailed values on OPV2V.

| Model | Maximum Latency ($t_{\text{lag.}}$) | | | | | |
|---|---|---|---|---|---|---|
| | 0 ms | 100 ms | 200 ms | 300 ms | 400 ms | 500 ms |
| CoBEVT | 0.659 | 0.656 | 0.646 | 0.630 | 0.612 | 0.593 |
| V2X-ViT | 0.685 | 0.682 | 0.677 | 0.670 | 0.664 | 0.658 |
| Where2comm | 0.675 | 0.673 | 0.661 | 0.647 | 0.632 | 0.621 |
| CoAlign | 0.638 | 0.634 | 0.624 | 0.614 | 0.603 | 0.593 |
| ParCon (Ours) | **0.713** | **0.711** | **0.705** | **0.697** | **0.690** | **0.683** |
| | $0.0°$ | $0.2°$ | $0.4°$ | $0.6°$ | $0.8°$ | $1.0°$ |
| CoBEVT | 0.659 | 0.656 | 0.646 | 0.630 | 0.612 | 0.593 |
| V2X-ViT | 0.685 | 0.682 | 0.677 | 0.670 | 0.664 | 0.658 |
| Where2comm | 0.675 | 0.673 | 0.661 | 0.647 | 0.632 | 0.621 |
| CoAlign | 0.638 | 0.634 | 0.624 | 0.614 | 0.603 | 0.593 |
| ParCon (Ours) | **0.713** | **0.711** | **0.705** | **0.697** | **0.690** | **0.683** |
| Model | Localization Error std. ($\sigma_{\text{hdg.}}$) | | | | | |
| | 0.0 m | 0.1 m | 0.2 m | 0.3 m | 0.4 m | 0.5 m |
| CoBEVT | 0.659 | 0.639 | 0.575 | 0.490 | 0.416 | 0.351 |
| V2X-ViT | 0.685 | 0.679 | 0.575 | 0.490 | 0.416 | 0.351 |
| Where2comm | 0.675 | 0.661 | 0.615 | 0.569 | 0.536 | 0.510 |
| CoAlign | 0.638 | 0.621 | 0.569 | 0.517 | 0.480 | 0.456 |
| ParCon (Ours) | **0.713** | **0.706** | **0.688** | **0.663** | **0.639** | **0.618** |

Table 14: Robustness to various ranges of noises with detailed values on DAIR-V2X.

| Model | Heading Noise std. ($\sigma_{\text{loc.}}$) | | | | | |
|---|---|---|---|---|---|---|
| | $0.0°$ | $0.2°$ | $0.4°$ | $0.6°$ | $0.8°$ | $1.0°$ |
| CoBEVT | 0.448 | 0.446 | 0.440 | 0.433 | 0.427 | 0.421 |
| V2X-ViT | 0.449 | 0.447 | 0.441 | 0.434 | 0.428 | 0.423 |
| Where2comm | 0.404 | 0.404 | 0.400 | 0.396 | 0.391 | 0.387 |
| CoAlign | 0.408 | 0.407 | 0.403 | 0.399 | 0.394 | 0.391 |
| ParCon (Ours) | **0.466** | **0.464** | **0.458** | **0.452** | **0.445** | **0.440** |
| Model | Localization Error std. ($\sigma_{\text{hdg.}}$) | | | | | |
| | 0.0 m | 0.1 m | 0.2 m | 0.3 m | 0.4 m | 0.5 m |
| CoBEVT | 0.448 | 0.439 | 0.424 | 0.408 | 0.397 | 0.388 |
| V2X-ViT | 0.449 | 0.439 | 0.426 | 0.411 | 0.401 | 0.394 |
| Where2comm | 0.404 | 0.398 | 0.387 | 0.376 | 0.368 | 0.363 |
| CoAlign | 0.408 | 0.402 | 0.393 | 0.384 | 0.376 | 0.371 |
| ParCon (Ours) | **0.466** | **0.458** | **0.441** | **0.426** | **0.414** | **0.406** |

Table 15: Robustness to a subset of noises with detailed values on V2XSet.

| Model | Maximum Latency ($t_{\text{lag.}}$) | | |
| --- | --- | --- | --- |
| | 0 ms | [100 ms, 200 ms] | [300 ms, 500 ms] |
| CoBEVT | **0.764** | 0.748 (2.09% ↓) | 0.697 (8.77% ↓) |
| V2X-ViT | 0.747 | 0.730 (2.24% ↓) | 0.686 (8.16% ↓) |
| Where2comm | 0.741 | 0.731 (1.35% ↓) | 0.688 (7.21% ↓) |
| CoAlign | 0.718 | 0.703 (2.09% ↓) | 0.671 (6.54% ↓) |
| ParCon (Ours) | 0.760 | **0.760 (0.03% ↓)** | **0.729 (4.01% ↓)** |

| Model | Heading Noise std. ($\sigma_{\text{hdg.}}$) | | |
| --- | --- | --- | --- |
| | 0.0° | [0.2°, 0.4°] | [0.6°, 1.0°] |
| CoBEVT | **0.764** | **0.763 (0.17% ↓)** | 0.735 (3.79% ↓) |
| V2X-ViT | 0.747 | 0.746 (**0.05% ↓**) | 0.730 (2.17% ↓) |
| Where2comm | 0.741 | 0.735 (0.79% ↓) | 0.710 (4.16% ↓) |
| CoAlign | 0.718 | 0.714 (0.47% ↓) | 0.698 (2.75% ↓) |
| ParCon (Ours) | 0.760 | 0.759 (0.18% ↓) | **0.744 (2.13% ↓)** |

| Model | Localization Noise std. ($\sigma_{\text{loc.}}$) | | |
| --- | --- | --- | --- |
| | 0.0 m | [0.1 ms, 0.2 ms] | [0.3 ms, 0.5 ms] |
| CoBEVT | **0.764** | 0.732 (4.26% ↓) | 0.548 (28.28% ↓) |
| V2X-ViT | 0.747 | 0.727 (2.58% ↓) | 0.669 (10.43% ↓) |
| Where2comm | 0.741 | 0.715 (3.61% ↓) | 0.631 (14.82% ↓) |
| CoAlign | 0.718 | 0.687 (4.26% ↓) | 0.601 (16.20% ↓) |
| ParCon (Ours) | 0.760 | **0.746 (1.81% ↓)** | **0.701 (7.75% ↓)** |

Table 16: Robustness to a subset of noises with detailed values on OPV2V.

| Model | Maximum Latency ($t_{\text{lag.}}$) | | |
| --- | --- | --- | --- |
| | 0 ms | [100 ms, 200 ms] | [300 ms, 500 ms] |
| CoBEVT | 0.659 | 0.644 (2.33% ↓) | 0.610 (7.43% ↓) |
| V2X-ViT | 0.685 | 0.662 (3.37% ↓) | 0.636 (7.13% ↓) |
| Where2comm | 0.675 | 0.661 (2.15% ↓) | 0.642 (4.97% ↓) |
| CoAlign | 0.638 | 0.626 (1.99% ↓) | 0.611 (4.26% ↓) |
| ParCon (Ours) | **0.713** | **0.705 (1.23% ↓)** | **0.689 (3.38% ↓)** |

| Model | Heading Noise std. ($\sigma_{\text{hdg.}}$) | | |
| --- | --- | --- | --- |
| | $0.0°$ | $[0.2°, 0.4°]$ | $[0.6°, 1.0°]$ |
| CoBEVT | 0.659 | 0.651 (1.24% ↓) | 0.612 (7.17% ↓) |
| V2X-ViT | 0.685 | 0.679 (0.85% ↓) | 0.664 (**3.06%** ↓) |
| Where2comm | 0.675 | 0.667 (1.27% ↓) | 0.633 (6.28% ↓) |
| CoAlign | 0.638 | 0.629 (1.45% ↓) | 0.603 (5.52% ↓) |
| ParCon (Ours) | **0.713** | **0.708 (0.77% ↓)** | **0.690** (3.32% ↓) |

| Model | Localization Noise std. ($\sigma_{\text{loc.}}$) | | |
| --- | --- | --- | --- |
| | 0.0 m | [0.1 ms, 0.2 ms] | [0.3 ms, 0.5 ms] |
| CoBEVT | 0.659 | 0.607 (7.89% ↓) | 0.419 (36.40% ↓) |
| V2X-ViT | 0.685 | 0.670 (**2.24%** ↓) | 0.608 (11.30% ↓) |
| Where2comm | 0.675 | 0.638 (5.53% ↓) | 0.538 (20.32% ↓) |
| CoAlign | 0.638 | 0.595 (6.74% ↓) | 0.484 (24.12% ↓) |
| ParCon (Ours) | **0.713** | **0.697** (2.28% ↓) | **0.640 (10.29% ↓)** |

Table 17: Robustness to a subset of noises with detailed values on DAIR-V2X.

| Model | Heading Noise std. ($\sigma_{\text{hdg.}}$) | | |
| --- | --- | --- | --- |
| | $0.0°$ | $[0.2°, 0.4°]$ | $[0.6°, 1.0°]$ |
| CoBEVT | 0.448 | 0.443 (1.01% ↓) | 0.427 (4.56% ↓) |
| V2X-ViT | 0.449 | 0.444 (1.04% ↓) | 0.429 (4.49% ↓) |
| Where2comm | 0.404 | 0.402 (**0.50%** ↓) | 0.391 (**3.12%** ↓) |
| CoAlign | 0.408 | 0.405 (0.78% ↓) | 0.394 (3.37% ↓) |
| ParCon (Ours) | **0.466** | **0.461** (0.92% ↓) | **0.446** (4.27% ↓) |

| Model | Localization Noise std. ($\sigma_{\text{loc.}}$) | | |
| --- | --- | --- | --- |
| | 0.0 m | [0.1 ms, 0.2 ms] | [0.3 ms, 0.5 ms] |
| CoBEVT | 0.448 | 0.431 (3.68% ↓) | 0.398 (11.13% ↓) |
| V2X-ViT | 0.449 | 0.432 (3.62% ↓) | 0.402 (10.37% ↓) |
| Where2comm | 0.404 | 0.392 (2.89% ↓) | 0.369 (8.63% ↓) |
| CoAlign | 0.408 | 0.398 (**2.59%** ↓) | 0.377 (**7.34%** ↓) |
| ParCon (Ours) | **0.466** | **0.449** (3.49% ↓) | **0.415** (10.75% ↓) |

Table 18: Comparison robustness to various ranges of noises between ParCon-S and ParCon on V2XSet.

| Model | Maximum Latency ($t_{\text{lag.}}$) | | | | | |
|---|---|---|---|---|---|---|
| | 0 ms | 100 ms | 200 ms | 300 ms | 400 ms | 500 ms |
| ParCon-S | 0.752 | 0.757 | 0.746 | 0.728 | 0.718 | 0.711 |
| ParCon (Ours) | **0.760** | **0.765** | **0.754** | **0.737** | **0.729** | **0.723** |

| Model | Heading Noise std. ($\sigma_{\text{loc.}}$) | | | | | |
|---|---|---|---|---|---|---|
| | $0.0°$ | $0.2°$ | $0.4°$ | $0.6°$ | $0.8°$ | $1.0°$ |
| ParCon-S | 0.752 | 0.752 | 0.748 | 0.741 | 0.734 | 0.726 |
| ParCon (Ours) | **0.760** | **0.760** | **0.757** | **0.751** | **0.744** | **0.737** |

| Model | Localization Error std. ($\sigma_{\text{hdg.}}$) | | | | | |
|---|---|---|---|---|---|---|
| | 0.0 m | 0.1 m | 0.2 m | 0.3 m | 0.4 m | 0.5 m |
| ParCon-S | 0.752 | 0.744 | 0.729 | 0.709 | 0.700 | 0.683 |
| ParCon (Ours) | **0.760** | **0.754** | **0.738** | **0.719** | **0.700** | **0.683** |

Table 19: Comparison robustness to a subset of noises between ParCon-S and ParCon with detailed values on V2XSet.

| Model | Maximum Latency ($t_{\text{lag.}}$) | | |
|---|---|---|---|
| | 0 ms | [100 ms, 200 ms] | [300 ms, 500 ms] |
| ParCon-S | 0.752 | 0.752 (0.05% ↓) | 0.719 (4.38% ↓) |
| ParCon (Ours) | **0.760** | **0.760 (0.03% ↓)** | **0.729 (4.01% ↓)** |

| Model | Heading Noise std. ($\sigma_{\text{hdg.}}$) | | |
|---|---|---|---|
| | $0.0°$ | $[0.2°, 0.4°]$ | $[0.6°, 1.0°]$ |
| ParCon-S | 0.752 | 0.750 (0.28% ↓) | 0.734 (2.45% ↓) |
| ParCon (Ours) | **0.760** | **0.759 (0.18% ↓)** | **0.744 (2.13% ↓)** |

| Model | Localization Noise std. ($\sigma_{\text{loc.}}$) | | |
|---|---|---|---|
| | 0.0 m | [0.1 ms, 0.2 ms] | [0.3 ms, 0.5 ms] |
| ParCon-S | 0.752 | 0.737 (2.05% ↓) | 0.689 (8.43% ↓) |
| ParCon (Ours) | **0.760** | **0.746 (1.81% ↓)** | **0.701 (7.75% ↓)** |