# OpenReview forum: "ParCon: Noise-Robust Collaborative Perception via Multi-module Parallel Connection"
_ICLR.cc/2025/Conference — ICLR 2025 Conference Withdrawn Submission_

### Official Review · Reviewer_2WLe · 2024-11-01

**Soundness:** 1
**Presentation:** 2
**Contribution:** 1
**Rating:** 3
**Confidence:** 5

**Summary:**

This paper presents a method for addressing the noise in cooperative perception via the proposed multi-module parallel connection. The proposed method employs a parallel architecture instead of sequential processing.

**Strengths:**

* The author benchmarked the performance on multiple datasets including simulation dataset (e.g., OPV2V, V2XSet) and real-world dataset (e.g., DAIR-V2X)

**Weaknesses:**

* Lack of novelty: The paper presents a method closely related to V2X-ViT with two main modifications: a parallel architecture for multi-agent and spatial attention instead of a sequential setup, and a replacement of window attention with a CNN-based S-Conv and Dilated Neighborhood Attention Transformer for S-Att. However, the novelty and justification for these modifications appear limited, and key design choices lack sufficient exploration.
* Motivation is insufficient: The motivation for using a parallel rather than sequential architecture for better robustness against noise for perception module is insufficient, with no clear rationale for improved noise handling or scalability.
* The tested heading angle range is narrow, making it unrealistic for real-world applications. Please consider increasing the experimented value range for more realistic setups.

**Questions:**

Clarification on Table 8: The meaning of "Original" is unclear. If it refers to a sequential setup, the performance gain might merely result from the larger model size. This distinction is essential for whether the performance gain is from the parallel design or purely from the extra model size.

---

### Official Review · Reviewer_Pk52 · 2024-11-04

**Soundness:** 3
**Presentation:** 3
**Contribution:** 2
**Rating:** 5
**Confidence:** 4

**Summary:**

This paper presents a new collaborative perception architecture that replaces the previous sequential module connections with parallel ones. Experiments demonstrate the robustness of this parallel architecture, and the proposed connections are also computationally efficient.

**Strengths:**

1. The writing is clear and easy to follow.
2. The motivation and proposed method are clearly stated.
3. This work introduces parallel module connections for collaborative perception.
4. The experiments are thorough, encompassing three datasets that include both real-world and simulated data, as well as addressing two practical issues: communication latency and pose errors, along with detailed ablation studies on the modules.
5. It is encouraging about the effectiveness, efficiency and robustness of the proposed method.

**Weaknesses:**

1. Lack of novelty and deeper analysis regarding the design of parallel connections. The different modules utilize various operators for information extraction, resembling an ensemble. Could you conduct ablation studies comparing their parallel architecture to sequential and ensemble approaches? Should the parallel designs be more diverse to complement each other better for improved performance? What types of operators are necessary for feature extraction in the context of collaborative perception? Could you analyze which specific types of features or information each module extracts, and how these complement each other for collaborative perception tasks?
2. There is confusion regarding the effectiveness of S-Att. The ablation results in Table 8 indicate that the gain from S-Att is minimal, while Table 9 suggests its effectiveness is significant.  Could you explain this apparent discrepancy and provide additional analysis to clarify the true impact of S-Att? For example, could you conduct more detailed ablation studies or provide visualizations of the features extracted by S-Att compared to other modules?
3. There is insufficient analysis of the experimental results. The findings primarily demonstrate facts but lack an exploration of the underlying reasons for the observed outcomes. For instance, why do the designed modules and their parallel connections contribute to robustness? Could you visualize attention maps or feature activations to show how information flows through the parallel modules under different noise conditions?

**Questions:**

1. In Figure 6, what do the lines in Depth 3 represent?
2. Why does the No Fusion value (0.754/0.602) in Table 1 differ from the Without A-Att value (0.774/0.645) in Table 8?

---

### Official Review · Reviewer_66ma · 2024-11-05

**Soundness:** 3
**Presentation:** 2
**Contribution:** 3
**Rating:** 5
**Confidence:** 3

**Summary:**

The paper introduces ParCon, a noise-robust collaborative perception model designed to improve 3D object detection in V2X (Vehicle-to-Everything) systems. ParCon introduce a concept for parallel  ollaborative perception, and propose to employ a parallel architecture that independently processes data streams from multiple agents, enhancing noise resilience and computational efficiency. With attention modules tailored to fuse different information types, such as agent-wise and spatial-wise data, ParCon achieves better overall accuracy under varying noise conditions and reduces FLOPs.

**Strengths:**

- ParCon proposes the adoption of a parallel architecture in V2X systems, allowing each module to process information independently and mitigate the propagation of noise.
- The framework is specifically designed for parallel fusion in V2X systems, incorporating attention mechanisms to fuse diverse types of information effectively, leading to improved detection accuracy.
- The model performs well across a variety of noise conditions, showcasing resilience and robustness compared to sequential models, particularly under latency, heading, and localization noise, with reduced FLOPs demonstrate the model efficiency.

**Weaknesses:**

- The paper appears to have been written hastily, with several typos, such as in the equations on lines 171 and 211. Many equations could be embedded within paragraphs for improved readability.
- Some descriptions, like the difference between sub-manifold sparse convolution and sparse convolution, lack clarity, making it challenging to understand the distinctions and implications.

**Questions:**

- Is this the first parallel collaborative perception model for V2X systems? If so, emphasizing this novelty in the paper could strengthen its contribution.
- How does ParCon manage relative pose errors? Communication errors often propagate differently across agents; addressing this could enhance robustness.
- Could you provide inference runtime metrics rather than only FLOPS to better understand the model’s real-world applicability?
- ParCon’s performance on the DAIR-V2X dataset under moderate noise conditions is inconsistent with other metrics, and the improvement via introducing s-att seems trival. Could the author explain more about the experimental results?

---

### Official Review · Reviewer_HgbP · 2024-11-11

**Soundness:** 2
**Presentation:** 1
**Contribution:** 1
**Rating:** 3
**Confidence:** 4

**Summary:**

The paper introduces ParCon, a collaborative perception model for autonomous vehicles that employs a parallel connection architecture instead of the sequential connections used in existing methods. The main goal of ParCon is to enhance perception accuracy and robustness in noisy communication environments, such as those encountered in V2X (Vehicle-to-Everything) systems. The authors present ablation studies to validate the effectiveness of individual components, such as CCL and parallel connections, and provide qualitative visualizations to demonstrate the advantages of their model in noisy and occluded environments. They conclude with a discussion of the limitations and propose future directions for enhancing the architecture with advanced sub-modules.

**Strengths:**

The paper introduces a parallel connection architecture for collaborative perception, addressing the challenge of noise robustness in V2X systems. It demonstrates improved detection accuracy and computational efficiency compared to state-of-the-art methods through the use of a Channel Compression Layer (CCL) and spatial-wise attention. The evaluation spans multiple datasets and provides empirical evidence of robustness under various noise conditions, making a practical contribution to autonomous driving research.

**Weaknesses:**

1. Writing: The use of the term "module" is vague and inconsistent throughout the paper. Does it refer to a computational block, a neural network component, or a specific feature processing unit? This ambiguity undermines the reader's understanding of the architecture.
Phrases like "manages noise independently" and "compensates limitations" lack specificity. How the proposed modules achieve these outcomes is poorly explained, leaving the mechanism unclear. The writing is repetitive and verbose, making it difficult to parse the main contributions and technical details. For example, the description of the Channel Compression Layer (CCL) is overly long, yet still leaves the reader unclear about its exact role. Key claims, such as computational efficiency and noise robustness, are repeated multiple times without sufficient empirical backing or theoretical analysis.

2. Novelty: The primary claim of novelty, a "parallel connection" architecture, is incremental and not supported by any theoretical insights or innovative algorithmic developments. This feature feels more like an engineering optimization rather than a fundamental research contribution. Many components of the architecture (e.g., PointPillars, Sparse ResNet, attention mechanisms) are off-the-shelf methods, and the way they are combined lacks originality. The authors fail to show how these combinations lead to insights or advancements that are relevant to ICLR.

3. Relevance: The paper’s focus on a specific application (collaborative perception for autonomous vehicles) lacks sufficient theoretical contributions or broader applicability to the machine learning community. The proposed method is narrowly tailored to V2X scenarios, which limits its relevance for a broader audience. The paper does not introduce new machine learning techniques or paradigms but rather applies existing concepts in a slightly modified architecture.

4. Related works: The paper could benefit from a more comprehensive literature review, as some highly relevant works about collaborative perception are missing [1-6]. Including a broader range of recent studies would provide a stronger context for the contributions and better situate the proposed framework within the current state of research.

[1] Li, Y., Ren, S., Wu, P., Chen, S., Feng, C. and Zhang, W., 2021. Learning distilled collaboration graph for multi-agent perception. Advances in Neural Information Processing Systems, 34, pp.29541-29552.

[2] Li, Y., Ma, D., An, Z., Wang, Z., Zhong, Y., Chen, S. and Feng, C., 2022. V2X-Sim: Multi-agent collaborative perception dataset and benchmark for autonomous driving. IEEE Robotics and Automation Letters, 7(4), pp.10914-10921.

[3] Huang, S., Zhang, J., Li, Y. and Feng, C., 2024. Actformer: Scalable collaborative perception via active queries. ICRA 2024.

[4] Yang, D., Yang, K., Wang, Y., Liu, J., Xu, Z., Yin, R., Zhai, P. and Zhang, L., 2024. How2comm: Communication-efficient and collaboration-pragmatic multi-agent perception. Advances in Neural Information Processing Systems, 36.

[5] Su, S., Li, Y., He, S., Han, S., Feng, C., Ding, C. and Miao, F., 2023, May. Uncertainty quantification of collaborative detection for self-driving. In 2023 IEEE International Conference on Robotics and Automation (ICRA) (pp. 5588-5594). IEEE.

[6] Su, S., Han, S., Li, Y., Zhang, Z., Feng, C., Ding, C. and Miao, F., 2024. Collaborative multi-object tracking with conformal uncertainty propagation. IEEE Robotics and Automation Letters.

**Questions:**

The term "module" is used extensively but inconsistently. Can the authors provide a precise definition of what constitutes a "module" in their architecture? Does it refer to distinct functional units, neural network blocks, or something else? How are the modules in the proposed parallel connection fundamentally different from the components in sequential architectures?

---

### Note · Authors · 2024-11-14

I have read and agree with the venue's withdrawal policy on behalf of myself and my co-authors.